# The Potent Phytoestrogen 8-Prenylnaringenin: A Friend or a Foe?

**DOI:** 10.3390/ijms23063168

**Published:** 2022-03-15

**Authors:** Raimo Pohjanvirta, Atefeh Nasri

**Affiliations:** 1Department of Food Hygiene and Environmental Health, University of Helsinki, 00790 Helsinki, Finland; 2Department of Veterinary Biomedical Science, Western College of Veterinary Medicine, Saskatoon, SK S7N 5B4, Canada; nasriatefeh@gmail.com

**Keywords:** phytoestrogens, natural compounds, xanthohumol, isoxanthohumol, naringenin, flavonoids, beer

## Abstract

8-prenylnaringenin (8-PN) is a prenylated flavonoid, occurring, in particular, in hop, but also in other plants. It has proven to be one of the most potent phytoestrogens in vitro known to date, and in the past 20 years, research has unveiled new effects triggered by it in biological systems. These findings have aroused the hopes, expectations, and enthusiasm of a “wonder-drug” for a host of human diseases. However, the majority of 8-PN effects require such high concentrations that they cannot be reached by normal dietary exposure, only pharmacologically; thus, adverse impacts may also emerge. Here, we provide a comprehensive and up-to-date review on this fascinating compound, with special reference to the range of beneficial and untoward health consequences that may ensue from exposure to it.

## 1. Introduction

Hop (*Humulus lupulus L.*) has been used as a preservative and flavoring agent in beer for centuries. This plant is rich in phenolic acids, flavonoids, proanthocyanidins, prenylated chalcones, and flavanones, as well as catechins with potential therapeutic applications [1]. Historically, menstrual disturbances were commonly reported among female hop pickers [2], and hop extract was reported to be efficient in reducing menopause-associated hot flushes in women. Moreover, hop baths have been used for treatment of gynecological disorders over the years [3]. Therefore, a recurring suggestion was that hop may contain compounds with powerful estrogenic activities. The estrogenicity of hops was attributed to xanthohumol (XN) without any reported scientific data [2]. Preliminary studies on estrogenic potency of hop showed contradictory results: some studies revealed high estrogenic activity [4,5], while others found no or low estrogenic activity of hop [6]. These discrepancies might have been due to the varieties of extracts and the nature of assays used [5]. In 1999, Milligan et al. reported of a novel compound, 8-prenylnaringenin (8-PN), after a successful bioassay-guided fractionation of hop extracts, and provided the first evidence of its prominent estrogenic activity [7]. In fact, this compound had been originally identified from hops as early as 1984 [8] but had been forgotten thereafter. Since Milligan et al.’s discovery, numerous studies have substantiated the high estrogenic potency of 8-PN. With accumulating evidence on the wide exposure of 8-PN to the general population, and of its exceptionally high potency in vitro, concerns have also arisen regarding the possible adverse effects to humans. This review therefore presents a comprehensive and up-to-date view on 8-PN.

## 2. Occurrence, Sources of, and Exposure to 8-PN

Throughout the years, a wide variety of methods, predominantly mass spectrometry-based, have been developed for reliable quantitative determination of 8-PN and related prenylated flavonoids (Table 1). They have enabled the analysis of these substances in diverse matrices and revealed that the main source of human 8-PN exposure was beer drinking when hop was used during the brewing process [9]. Hops are added to beer in the form of dried hop or lipophilic extracts [10]. Hop extracts or dried hops are obtained from female hop cons that are particularly rich in prenylated chalcones (XN, desmethyl-XN) and prenylated flavanones (6-PN, 8-PN) [11]. The exact amount of each ingredient varies, because chalcones undergo isomerization during the brewing process, which results in the conversion of XN to isoxanthohumol (IX) and of desmethyl-XN to a racemic mixture 6- and 8-PNs (Figure 1). Moreover, the cyclization of XN to IX may generate two enantiomers of IX, which can lead to two enantiomers of 8-PN [12]. The concentration of XN may amount to 1% while 8-PN is more than 10 times less abundant [13].

The volume of beer consumed per capita varies substantially among countries, being highest in the Czech Republic (191.8 L in 2018) [14]. The concentration of 8-PN in beer also varies. For example, in a study reported by Stevens and Page (2004) [9], European lager was found to contain a mere 1 µg/L, whereas American porter possessed a concentration of 240 µg/L. Still, in people harboring high intestinal and/or hepatic biotransformation capacities (see 8-PN pharmacokinetics), the ultimate source of 8-PN is, in most cases, IX, because its concentration in beer exceeds that of 8-PN by 10–40-fold [9,15,16]. Assuming the consumption of 1 L of, for example, “imported stout” beer with 8-PN and IX concentrations of 69 and 2100 µg/L, respectively [15], a person endowed with efficient conversion capability (80%; [17]) would be exposed to ~1700 µg 8-PN. Possemiers et al. (2006) have also estimated that moderate beer consumption could lead to 8-PN exposure in the range of 1–2 mg/day [17].

In an opened beer bottle, IX concentration may decrease by almost 20% in a day and over 30% in a week. For 8-PN, these reductions can be even larger (up to 50% in a week) [18]. No differences in 8-PN, IX, or XN concentrations between organically and conventionally produced hops and beer were detected [19].

Apart from hops, the occurrence of 8-PN has been reported in at least 21 ethnomedicinally applied plants [20], including citrus plants [21]. It may also occur in herb or hop teas at low concentrations (<10 µg/L); IX concentrations seem to be only slightly higher in these products (<15 µg/L) [15,16].

Clarke and coworkers [22] identified and quantified phytoestrogens in 35 dietary supplements on sale in the UK, Canada, and Italy. None of the samples contained measurable levels of 8-PN, 6-PN, or 6,8-diprenylnaringenin. In a more recent study, Schretter et al. (2020) [23] were unable to detect 8-PN in two out of the three hop dietary supplements analyzed. In contrast, the third product would provide 144 µg 8-PN and 1740 µg IX at the recommended daily dosage, indicating huge differences among these products. Coldham and Sauer (2001) reported that two dietary supplements intended for breast enhancement contained ~10 µg/g 8-PN. This would result in a calculated daily intake of 103–155 µg [24].

In the study by Schretter et al. (2009) [23], a herbal medicinal product and two batches of the herbal drug “Lupuli flowers” (in pharmacopeia quality) proved to encompass similar quantities of IX and 8-PN, and the recommended daily dose for the medicinal product would lead to the intake of ~45 µg 8-PN and ~255 µg IX. Dhooghe et al. (2010) studied hop-derived capsules, tablets, and drops on the market in Belgium for their 8-PN content [25]. In two products, 8-PN levels were undetectable. At their recommended daily doses, the others would release from 40 to 150 µg 8-PN. Thus, 8-PN intake from hop dietary supplements or medicinal products can mostly amount to a level comparable to the exposure of beer-lovers with fairly poor IX conversion capability.

## 3. Methods Applied in 8-PN Identification and Quantification

The bulk of the methods described to date for 8-PN quantification have combined LC or HPLC separation with detection by mass spectrometry (MS), although a gas chromatography (GC)–MS method has also been published [15,19,20,26,27,28,29] (Table 1). The varieties of MS used include electron impact (EI), atmospheric pressure electrospray ionization (ESI), or atmospheric pressure chemical ionization (APCI) in either positive or negative ion modes [15,18,24,30]. Apart from MS, HPLC methods with ultraviolet (UV) or UV/diode array detector (DAD) detection have been developed [18,25,31,32]. Some modifications of these techniques include ultra-high-pressure LC–tandem MS (UHPLC–MS/MS), ultra-high-performance supercritical fluid chromatography (UHPSFC; with either UV or MS detection), and the application of secondary standards in HPLC–UV/DAD [23,25]. Antibody-based methods have also found their way to 8-PN quantification. Polyclonal antibodies were used for the design of a radioimmunoassay (RIA), and monoclonal for devising an enzyme-linked immunosorbent assay (ELISA) [33,34]. The most recent new technique described for 8-PN quantification is stable isotope dilution analysis (SIDA) [16].

In most cases, these quantitative methods have measured racemic 8-PN, but there are also a couple of enantiospecific analytic techniques among them [20,32]. Moreover, as a facile means to determine the enantiomeric purity of 8-PN, a method was devised in which diastereomeric (*1S*)-(–)-camphanic acid esters of the enantiomers are first produced to be subsequently separated by reversed-phase HPLC [35].

**Table 1 ijms-23-03168-t001:** Methods used in quantitative 8-PN analysis.

		Sensitivity		Precision		Accuracy		
Method ^1^	Matrix	LLOQ ^2^ (ng/mL)	LOD ^3^ (ng/mL)	CV ^4^-Intra (%)	CV-Inter (%)	RE ^5^ (%)	Related Analytes Measured	Reference
GC/MS-SIM	Beer, hop pellets	NP	5 (beer)	NP	NP	≤65 (beer)	–	[26]
LC–ESI–MS	Beer	2.4	0.8	2.0	8.0	8.8	–	[27]
LC–ESI–MS	Serum, urine ^6^	S: 50; U: 10	NP	S: ≤10.9; U: ≤14.9	S: ≤13.7; U: ≤14.1	S: 2.6; U: 2.2	–	[20]
LC−ESI-MS/MS	Urine, beer	5 (urine)	0.03 (urine)	≤13.9 (urine)	≤12.6 (urine)	≤14.6 (urine)	X, IX	[29]
HPLC–MS/MS	Beer, hop extracts, herb teas	NP	NP	8.8 (beer)	8.2 (beer)	≤10 (beer)	XN, IX, 6-PN	[15]
UHPLC–MS/MS	Serum	1	NP	≤10.5	≤12.1	≤4	XN, IX, 6-PN	[28]
UHPLC–MS/MS	Beer, hop pellets	NP	NP	≤6	≤5	≤13	XN, IX	[19]
HPLC–APCI–MS	Dietary supplement for breast enhancement	NP	NP	6	2	3	XN, IX, 6-PN, 6,8-diPN	[24]
HPLC–APCI–MS	Serum, urine	S: 4.8; U: 1.2	S: 1.5; U: 0.4	S: 3.9; U: 6.0	S: 9.7; U: 14.4	S: 7.4; U: 10.1	XN, IX	[30]
HPLC–APCI–MS/MS	Beer, hop extracts, herb teas	NP	NP	≤7.9	≤8.2 (beer)	≤10	XN, IX, 6-PN	[15]
HPLC–APCI–MS	Beer, hop extracts	20	6	NP	NP	NP	XN, IX	[18]
HPLC–UV	Beer, hop extracts	100	30	NP	NP	NP	XN, IX	[18]
HPLC-UV	Beer, hops, hop pellets	30	10	<0.3	≤ 2.0	≤4.8	XN, IX	[32]
HPLC-UV/DAD	Hop cultivars and genotypes	3800	1000	NP	NP	3.2	XN, IX, 6-PN	[31]
HPLC–UV/DAD (+SS)	Hop extract and capsules	NP	860	<4.0	≤ 5.0	≤8.1	XN, IX, 6-PN	[25]
UHPSFC–UV	Hop dietary supplements, herbal products	100	60	0.0 ^7^	0.1 ^7^	≤7.7	XN, IX, 6-PN	[23]
UHPSFC–MS	Hop dietary supplements, herbal products	50	20	0.02 ^7^	0.1 ^7^	NP	XN, IX, 6-PN	[23]
RIA (polyclonal) ^8^	Beer, urine	NP	0.3 (urine)	<9 (urine)	<27 (urine)	≤36 (urine)	–	[33]
ELISA (monoclonal) ^9^	Serum, urine	17.1	4.4	S: 2.4; U: 0.7	S: 6.1; U: 7.2	S: 5.6; U: 5.1	X, IX	[34]
SIDA-LC-MS/MS	Beer, hop pellets, and tea	1.3 (beer)	0.32 (beer)	2.09 (beer)	8.2 (beer)	≤13 (beer)	XN, IX, 6-PN	[16]
^1^ Method abbreviations:	APCI, atmospheric pressure chemical ionization; DAD, diode-array detection; ELISA, enzyme-linked immunosorbent assay; ESI, electrospray ionization; GC, gas chromatography;							
	HPLC, high-performance liquid chromatography; LC, liquid chromatography; MS, mass spectrometry; RIA, radioimmunoassay; SIDA, stable isotope dilution analysis;							
	UHPSFC, ultrahigh-performance supercritical fluid chromatography; UV, ultraviolet light							
^2^ Lower limit of quantification								
^3^ Limit of detection								
^4^ Coefficient of variation								
^5^ Relative error								
^6^ For both enantiomers								
^7^ Based on peak area								
^8^ Both enantiomers detected; cross-reactivity with XN, IX and 6-PN ≤ 0.15%								
^9^ Both enantiomers detected; cross-reactivity with XN and IX < 0.01%, with 6-PN < 0.03%								

## 4. Pharmacokinetics of 8-PN

Studies with pure 8-PN on other aspects of its pharmacokinetics than biotransformation are scarce. After the ingestion of a single oral dose of 500 mg racemic 8-PN in 16 healthy young adults (8 women, 8 men), a maximum plasma concentration was attained at 1.6 h with large inter-individual (but not gender-related) differences in the height of the peak [36]. Interestingly, at an equimolar dose, 6-PN peaked later (2.3 h) and its area under the plasma concentration–time curve remained much smaller. When three groups of eight healthy postmenopausal women were treated orally with 50, 250, or 750 mg racemic 8-PN, the compound was rapidly absorbed, leading to maximum serum drug concentrations 1.0–1.5 h after administration [37]. Thereafter, drug serum concentrations decreased sharply, followed by a further increase in concentrations leading to a second peak, which occurred at 7–10 h. This was suggestive of marked enterohepatic recirculation, probably attributable to the prenyl group in 8-PN [36].

In agreement with these in vivo reports attesting to effective gastrointestinal (GI) absorption of 8-PN, in monolayers of the human intestinal epithelial cancer cell line, (Caco-2), 8-PN also exhibited good absorption, probably via passive diffusion [38]. In these cells, 8-PN was biotransformed into two glucuronides (predominantly 4′-O-glucuronide) and two sulfates, with metabolism reaching 57% by 4 h. Human liver microsomes generated a total of 12 metabolites of 8-PN, and biotransformation occurred on the prenyl group and the flavanone skeleton [39]. Two of the most abundant of the metabolites, (E)-8-(4’’-hydroxy isopentenyl)naringenin (8-PN-OH) and (E)-8-(4’’-oxoisopentenyl)naringenin (8-PN=O), exhibited estrogenic activity in vitro. Cytochromes P450 (CYPs), including CYP2C8 and CYP2C19, catalyzed the phase I reaction of alcohol formation in the prenyl side chain of 8-PN [19]. The alcohol derivatives of 8-PN produced in hepatocytes were subsequently conjugated to at least six glucuronides but apparently no sulfates. In contrast to the situation in enterocytes, 7-O-glucuronide isomer was found to be the most abundant conjugate (~80% of all). The enzymes responsible for glucuronidation of 8-PN were uridine 5’-diphospho-glucuronosyltransferase (UGT)1A1, UGT1A6, UGT1A8, and UGT1A9. Only 4% of 8-PN remained unmetabolized after a 4-h incubation with human hepatocytes [38].

A more recent study by Fang et al. (2019) [40] examined whether 8-PN biotransformation in human liver microsomes or in reactions catalyzed by 11 recombinant human UGTs was enantiomer-specific. Incubation of racemic 8-PN with human liver microsomes resulted in the formation of (*2R*)-8-PN-7-O-glucuronide and (*2S*)-8-PN-7-O-glucuronide in a ratio of 1.37:1.00. Except for UGT1A1, the UGTs produced more (*2R*)-8-PN-O-glucuronides than the corresponding (*2S*)-glucuronides from purified (*2R*)-8-PN and (*2S*)-8-PN. Therefore, the half-life of (*2S*)-8-PN is likely to be longer in humans.

8-PN may also be metabolized in the human GI tract, because the anaerobic intestinal bacterium *Eubacterium ramulus* isomerized and hydrogenated 8-PN into two chalcones, O-desmethyl-XN and O-desmethyl-*α,β*-dihydroxanthohumol [41]. In addition to this bacterium, numerous fungi have been shown to be able to transform 8-PN [42]. Besides an oxygenated metabolite, which was also reported as a minor derivative of 8-PN in human liver microsomes [39], glucosylated, acyl-glucosylated, and sulfated fungi-generated metabolites have been identified [42,43,44,45].

Human liver microsomes (but not Aroclor-induced rat liver microsomes [46]) are capable of converting IX to 8-PN via O-demethylation [47], catalyzed by CYP1A2 [48]. Since this reaction can also occur in the GI tract and, e.g., beer usually contains 30–40 times more IX than 8-PN [30], the exposure to 8-PN is critically dependent on IX abundance and metabolism. In vitro studies implied that in the human GI tract, the conversion from IX to 8-PN almost exclusively occurs in distal colon [17]. Fecal samples from female volunteers revealed that the fecal microbiota could be separated into high (8/51), moderate (11/51), and slow (32/51) IX converters, with a mean 8-PN production of 78.8, 48.5, and 6.9%, respectively [2]. Classification of humans based on urinary excretion of 8-PN yielded similar ratios [49]. The microbial bioactivation of IX was inhibited by coexposure to two other proestrogens, daidzein and secoisolariciresinol, both in vitro and in vivo (by 20.5% and 35.4%, respectively) [50]. The intestinal bacterium responsible for the O-demethylation of IX to 8-PN was identified to be *Eubacterium limosum* [51]. After strain selection of the bacterium, a conversion efficiency of 90% was achieved [51,52].

Upon ingestion of pure 8-PN or 6-PN by young adults, the cumulative urinary excretion over 24 h for both compounds was less than 1.5% of the administered dose [36]. In the study by Rad et al. (2006) [37], described above, 4.8–8.3% of dose was excreted by 48 h in urine, almost entirely as 8-PN conjugates. In feces, slightly less than 25% of the dose administered was recovered as free 8-PN by 48 h but it may be an underestimation due to technical problems encountered. It was unclear to what extent this free 8-PN stemmed from de-conjugated metabolites, as conjugates of 8-PN seem to be hydrolyzed by human intestinal microbiota [53]. In support of these findings, in menopausal women ingesting standardized extracts of spent hops the elimination half-life for 8-PN was estimated to be over 20 h [54]. Hence, excretion of 8-PN appears to be a relatively slow process, mainly due to enterohepatic recirculation of conjugated and de-conjugated metabolites. This suggests that although oral 8-PN undergoes extensive first-pass metabolism in the GI tract and liver, it can reach effective biological concentrations due to its low elimination rate, with a possibility of accumulation in target tissues. Therefore, more studies with longer follow-up times are needed to ascertain these kinetic aspects.

Simultaneous exposure to 8-PN may influence the kinetics of pharmaceuticals, because 8-PN proved to be a potent inhibitor of the ABCG2 efflux transporter [55]. Likewise, it was reported to inhibit two other ABC transporters, P-glycoprotein (ABCB1/P-gp) and multidrug resistance-associated protein 1(ABCC1/MRP1) [56]. Thus, 8-PN possesses the potential to interfere with the efflux of concomitantly consumed drug substrates for these transporters. 

Finally, all of the human studies have examined racemic 8-PN. In a preliminary experiment in rats, the *S*-form of 8-PN appeared to be excreted in urine at a slightly faster rate than *R*-8-PN after iv-injection of racemic 8-PN [20]. As mentioned above, indirect evidence suggests that in humans the converse may be true. On the other hand, it should be noted that back conversion to a racemic mixture might occur from both enantiomers [57].

## 5. Estrogenic Activity of 8-PN

### 5.1. In Vitro Studies

8-PN is one of the most potent phytoestrogens currently known. This has been confirmed by a variety of in vitro assays for estrogenic activity (Table 2), including yeast-based screens expressing the human estrogen receptor (ER) and a reporter gene (usually luciferase, β-galactosidase or chloramphenicol acetyltransferase) under the control of estrogen-responsive sequences (ERE) [5,46,58,59], human cell lines responsive to estrogenic stimulation [5,7,60,61], and ER binding [5,7,61,62]. Compared with established phytoestrogens, such as coumestrol, genistein, and daidzein, the estrogenic activity of 8-PN has generally proven to be 0.6–150, 8–150, and 50–1500-fold stronger, respectively [5,7,59,62] (but see ERβ-specific data below), while concomitantly 5–250 times weaker than that of 17β-estradiol (E2) [5,7,46,59,60,61,62,63,64]. The estrogenic activities of IX and 6-PN have been reported to be ~1% and <1% relative to 8-PN, respectively, while XN appears to be devoid of estrogenic properties [7]. Likewise, the parent compound of 8-PN, naringenin, shows only very weak estrogenicity [65,66].

The studies cited above have primarily measured ERα-mediated responses. Initial studies suggested that the two enantiomers of 8-PN exhibit similar estrogenic activities in vitro and show no preferential affinity to the two forms of ERs—ERα and ERβ [5]. However, this view was later contested by Schaefer et al. (2003) [63], who reported that 2*S*(−)8-PN shows moderately higher ER affinity and estrogenic activity in vitro and in vivo than 2*R*(+) 8-PN. Furthermore, in their hands the affinity of 8-PN for ERα was >2-fold higher than for ERβ measured by in vitro competitive binding assay and the estrogenic activity at ERα was >3.6-fold higher than at ERβ in a transactivation analysis. In support of this, Overk et al. (2005) reported a 3-fold higher IC_50_ for ERβ vs. ERα binding [61], and Helle et al. (2014) recorded a 3-fold greater relative affinity to ERα vs. ERβ for 8-PN relative to E2 [66]. On the other hand, while using yeast cells expressing human ERs, Bovee et al. (2004) found quite the opposite: the affinity of 8-PN for ERβ was 3 times as high as for ERα [67]. In any case, even the latter study confirmed that relative to E2, 8-PN shows higher potency at ERα vs. ERβ, and in this regard appears to conspicuously stand out from the bulk of other phytoestrogens, such as coumestrol, genistein, and daidzein, which are all clearly more potent at ERβ [67]. In fact, genistein and daidzein may bind to ERβ with an affinity 85 or 1,4 times as great as that of 8-PN, respectively [61]. They have also been reported to activate ERβ at lower concentrations than 8-PN [68].

In a battery of tests measuring estrogenicity in estrogen-sensitive MCF-7 cells, the maximal responses generated by 8-PN were similar to those generated by E2, implying that it is a full estrogen agonist. Moreover, no evidence was found for any estrogen antagonist action by 8-PN at concentrations of up to 10^−6^ M [62]. Likewise, results from a yeast-based estrogen bioassay indicated a purely agonistic action at both ER subtypes [69]. In silico modeling of ER binding further suggested that chain prenylation without an increase in molecular length, which enables 8-PN to fit into the hydrophobic pocket in the human ER, is responsible for the high agonist activity of 8-PN compared with naringenin [69]. However, conflicting data were obtained in rat N46A-B14 cells derived from raphe nuclei region of the brain (thus serotonergic in origin) and stably expressing a transgenic human ERβ along with an ERE–luciferase construct [68]. In this test system, co-treatment of 8-PN with E2 resulted in a slight but statistically significant inhibition of E2 impact, suggesting partial agonism as the mode of interaction for 8-PN with ERβ. The same research team used these rat cells also in a subsequent study aimed at comparing other phytoestrogens with E2 for their influence on the expression of selected estrogen-responsive genes [70]. In proliferative conditions, 8-PN at a 1000-fold concentration faithfully reproduced the pattern of E2-induced effects; in differentiating conditions, the responsiveness of the analyzed genes towards E2 and 8-PN treatments was almost completely lost. 

In a yeast assay designed to reveal agonistic and antagonistic properties of substances on human ERα and androgen receptor, 8-PN again proved to be a pure ERα agonist. It had no activity on the androgen receptor [71]. However, this latter outcome was contested by another group also using a yeast-based androgen receptor assay: here, 8-PN demonstrated anti-androgenic activities. Yet, when assessed in the same study by prostate specific antigen production in androgen receptor positive PC3(AR)_2_ cells, 8-PN failed to show any anti-androgenicity [72]. A subsequent study revealed the critical role of the prenyl group at C(8) for 8-PN ER subtype and action selectivity [73]. When the prenyl group was substituted with alkyl chains of varying lengths and branching patterns, the new 8-alkylnaringenins were found to span an activity spectrum ranging from full agonism all the way via partial agonism to antagonism. An intriguing example is 8-(2,2-dimethylpropyl)naringenin, which was a potent agonist of ERα, but pronounced antagonist of ERβ. Overall, the antagonist effect of the derivatives was more substantial on ERβ than ERα [73]. Taken together, the weight of evidence points to a solely agonist action for 8-PN at ERα. At ERβ or the androgen receptor, the issue is not equally clear and awaits further studies.

In addition to its role as a ligand-activated transcription factor, ER can also mediate rapid nongenomic actions, such as activation of various protein-kinase cascades. These are believed to involve membrane-associated ERs [74]. Intriguingly, while both E2 and 8-PN induced the association of ERα with c-Src tyrosine kinase in MCF-7 cells, only E2 did the same with PI3 kinase. This failure of 8-PN resulted in a rapid decline in cyclin D1 and E steady state protein levels and induction of apoptosis [75]. It should be noted that induction of these non-genomic ER actions required high, pharmacological concentrations of 8-PN (≥10 µM), but still the difference in the bioactive profiles of E2 and 8-PN in vitro is noteworthy in light of their somewhat distinctive effects in subchronic exposure in vivo (see below).

In a recent study employing the human endometrial adenocarcinoma cell line, which expresses both ERα and ERβ, Ishikawa cells, alkaline phosphatase (AP) activity was used as a measure of estrogenic action [76]. The researchers were interested in finding out whether the prenylated compounds in beer might have functional interactions with one another or with pesticides whose residues may occur in beer. They discovered that XN and IX exhibited from additive to slightly synergistic combinatory effect with 8-PN, and the mixture reached its peak induction of AP activity (about 80% of that caused by 1 nM E2) at a dose which corresponded to 10% of the estimated average dose in beer. A slightly synergistic interaction was also recorded between the mixture of 8-PN, IX, and XN and that of the pesticides. In contrast to this outcome, 8-PN did not modify (or subtly inhibited) the estrogenic activities of two fungal xenoestrogens, zearalenone and α-zearalenol in Ishikawa cells. XN antagonized the fungal products more substantially [77].

### 5.2. In Vivo Studies

In in vivo conditions, the estrogenicity of 8-PN may be less prominent than in vitro. In juvenile male medaka fish, the positive control compound ethinylestradiol at the concentration of 0.34 nM induced sex reversal as indicated by the expression of the ovary-specific gene *figla* and the estrogen-dependent gene *vitellogenin*. However, 674 nM 8-PN (i.e., ~2000-fold concentration) failed to cause either of these effects [64]. When assessed by uterine vascular responses in ovariectomized (OVX) mice after subcutaneous (sc.) administration, the estrogenic activity of 8-PN was only <1% compared with E2. It was even lower (<0.1%) based on mitotic activity of vaginal epithelia upon peroral treatment. Moreover, in contrast to E2, 8-PN failed to induce mitosis in uterine epithelium or increase uterus weight in OVX mice at oral doses of up to 16 mg/kg/day for 3 days. [5]. This may be a species-dependent failure, because in OVX Wistar rats treated for 3 days with 10 mg/kg/day of either 8-PN or genistein, 8-PN proved to cause a greater elevation in uterus weight (~2 times the control weight) than genistein (~1.4 times the control weight) [68]. Exactly the same dosing regimen for 8-PN and animal model was also applied in another study in which the aim was to compare the impacts of E2 and 8-PN in vivo [78]. E2 at 30 µg/kg/day increased uterine wet weight as well as epithelial height in the uterus and vagina and altered expression of estrogen-responsive genes in the uterus and liver. 8-PN elicited an identical pattern of responses, but the effects were blunted except for the induction of *insulin-like growth factor binding protein-1* (IGF-1) gene expression in the liver, which was more prominent upon 8-PN than E2 treatment.

Similarly, sc. exposure of young adult OVX Lewis rats to 15 mg/kg/day 8-PN for three days induced uterine weight 1.8-fold and elevated the height of the glandular epithelium [79]. However, in identical experimental conditions, E2 at the dose of 4 µg/kg/day increased uterine wet weight by 4-fold augmenting the height of the luminal and glandular epithelia as well as myometrial thickness. In the vagina, 8-PN failed to stimulate the epithelial thickness but increased the percentage of proliferation cell nuclear antigen (PCNA)-positive epithelial cells as quantified by immunohistochemistry, which implies that epithelial cell proliferation was promoted all the same. This research team also assessed by an identical experimental setting the effect of 8-PN on the mammary gland [66]. Here, 8-PN induced the formation of terminal end buds and stimulated expression of proliferation markers in epithelial ductal cells with the same pattern as 4 µg/kg/day E2, albeit in a much smaller magnitude [66]. The only exception was progesterone receptor, whose gene expression was induced by E2 alone.

A time-course study was conducted in OVX Wistar rats by exposing sc. the animals to 10 µg/kg/day of E2 or 15 mg/kg/day of 8-PN and harvesting the uteri at 7, 24, or 72 h [80]. E2 elevated uterus wet weight and modified the expression levels of estrogen-sensitive genes in this tissue. The temporal and directional patterns in these variables were highly similar after 8-PN administration. In another rat study with a single dose (amount not reported) to juvenile female animals, 8-PN proved to be 20,000 times weaker than E2 in its ability to stimulate the mass of uterus or vagina [63]. It is also relevant in this context that in a rat model of menopausal hot flashes, represented by an elevation in the tail skin temperature of estrogen-deprived (OVX) rats, a 6-day oral or sc. administration of 8-PN was able to restore the temperature into the normal range. The effect was similar to that caused by E2 at a 100-fold lower dose and blocked by an ER antagonist [81]. Overall, it seems that in qualitative terms, 8-PN closely resembles E2 in its profile of short-term in vivo effects but possesses a lower relative potency than in in vitro conditions, presumably due to its rapid metabolism.

The estrogenic effects of 8-PN have further been assessed after a longer exposure period. In OVX rats treated with 30 mg/kg/day 8-PN for two weeks, 8-PN was reported to increase uterus weight and also suppress ovariectomy-induced bone loss [82]. Treatment of adult OVX Sprague–Dawley rats with 4 or 40 mg/kg/day IP for 21 days increased uterus weight (although less than 10 µg/rat/day E2), while 0.4 mg/kg/day was ineffective [83]. The three 8-PN treatment levels dose-dependently enhanced uterine epithelial cell height, with the effect maximally amounting to about half of the impact of E2. In a subchronic study, OVX Sprague–Dawley rats were treated with either E2 (0.17 or 0.7 mg/kg/day) or 8-PN (6.8 or 68.4 mg/kg/day) added in their soy-free chow for three months, and uterine, vaginal, and mammary gland samples were analyzed histologically [84]. Compared with untreated OVX rats, the higher 8-PN dose augmented uterine weight (by increasing endo- and myometrial thickness), but the low and high E2 treatments were 1.7 and 2.0 times, respectively, as effective in this regard. The lower 8-PN dose did not morphologically affect the uterus, while the high doses of E2 and 8-PN led to the formation of squamous metaplasia, cystic glands, and hyperplastic/hypertrophic glands (the last ones were also detected in the low E2 group). Interestingly, two changes occurred exclusively in a single group. Forty-five percent of the rats in the high E2 group exhibited pyometra, whereas 60% of the high 8-PN rats showed polypoid structures with cysts. A clear distinction between the two compounds was also seen in the vagina. Whereas all E2- and no 8-PN-exposed rats displayed keratinization, vacuolization was only recorded in 8-PN groups (clear in 90% of the high dose group, incipient in 55% of the low and 10% of the high dose groups). In the mammary gland, there was luminal formation clearly observable in all animals treated with E2 and in 40% of the high-8-PN group. Thus, after subchronic exposure in rats, the estrogenic effect profile of 8-PN appears to be largely but not completely overlapping with E2 [84].

**Table 2 ijms-23-03168-t002:** Estrogenic activity of 8-PN.

1. In Vitro								
Test Assay	Type of 8-PN	Variable Measured	8-PN	E2	Coumestrol	Genistein	Daidzein	Reference
Binding to ER in rat uterine cytosol	Racemic	Relative affinity	0.023	1	0.008	0.003		[7]
Stimulation of alkaline phosphatase	Racemic	EC_50_ (nM)	4	0.8	30	200	1500	[7]
activity in Ishikawa cells								
ERE-reporter gene expression in yeast	Racemic	EC_50_ (nM)	40	0.3	70	1200	2200	[7]
cells transfected with the human ERα								
Human ERα binding in vitro	(*R*)-8-PN	Relative affinity	~0.01	1				[5]
Human ERα binding in vitro	(*S*)-8-PN	Relative affinity	~0.01	1				[5]
Human ERα binding in vitro	(*R*)-8-PN	Relative affinity	~0.01	1				[5]
Human ERα binding in vitro	(*S*)-8-PN	Relative affinity	~0.01	1				[5]
Inhibition of E2 binding to ER in MCF-7 cells	Racemic	Molar excess over E2	45		35	1000	>10^4^	[62]
ERE-CAT induction in MCF-7 cells	Racemic	Relative potency	100	1	330	250	3300	[62]
Proliferation of MCF-7 cells after 7 days	Racemic	Relative potency	3.3	1	500	500	5000	[62]
ERE-reporter gene expression in yeast	Racemic	EC_50_ (nM)	0.02	0.005		40		[85]
cells transfected with the human ERα								
Proliferation of MCF-7 cells after 24 h	Racemic	EC_50_ (nM)	5	0.2		830		[85]
ERE-reporter gene expression in yeast	Racemic	EC_50_ (nM)	100	0.8	140	2000	No resp.	[67]
cells transfected with the human ERα								
ERE-reporter gene expression in yeast	Racemic	EC_50_ (nM)	33	0.2	3	8	800	[67]
cells transfected with the human ERβ								
Human ERα binding in vitro	Racemic	IC_50_ (nM)	510	20		300	17,000	[61]
Human ERβ binding in vitro	Racemic	IC_50_ (nM)	1700	15		20	1200	[61]
Human ERα binding in vitro	Racemic	IC_50_ (nM)	59	11				[66]
Human ERβ binding in vitro	Racemic	IC_50_ (nM)	65	4.2				[66]
ERE-reporter gene expression in yeast	Racemic	EC_50_ (nM)	130	0.8		9300		[64]
cells transfected with the human ERα								
Binding to the ligand-binding domain	Racemic	IC_50_ (nM)	57	1.2		1145		[73]
of human ERα in vitro								
Binding to the ligand-binding domain	Racemic	IC_50_ (nM)	68	1.4		25		[73]
of human ERβ in vitro								
**2. In vivo**								
**Assay used**	**Type of 8-PN**	**Variable measured**	**8-PN**	**E2**	**Coumestrol**	**Genistein**	**Daidzein**	**Reference**
Vascular permeability in mouse uterus	Racemic	Relative potency	<0.01	1	<0.01	<0.001	No effect	[5]
Vaginal epithelial mitosis	Racemic	Relative potency	<0.001	1				[5]
in OVX mice								

## 6. Endocrine Roles of Estrogen and 8-PN

Secondary metabolites from plants, including 8-PN, have been considered as alternatives to the classic hormone therapy in women. Studies on the endocrine properties of 8-PN have demonstrated that this compound is a natural selective estrogen receptor modulator (SERM) because its effect spectrum is not fully identical with that of E2. As already referred to earlier, E2 induces cellular changes through nuclear and non-nuclear mechanisms, and ER exists in 2 forms, ERα and ERβ, which have multiple isoforms and exhibit distinct tissue expression patterns and functions [86,87]. The classical nuclear mechanism of ER’s action typically occurs within hours, leading to activation or repression of target genes. However, estrogens can also induce rapid signals that act within seconds or minutes through extranuclear, membrane-associated forms of ERs as well as a G protein-coupled estrogen receptor (GPCR1; also called GPCR30) [88]. 

Despite the similar affinity of the two well-established SERMs tamoxifen and endoxifen to both isoforms of ERs [89], SERMs usually exhibit preference to one isoform of ER over the other one, which may explain the varying and tissue-specific responses to SERMs. As described earlier, the majority of phytoestrogens, inclusive of coumestrol, genistein, apigenin, naringenin, and kaempferol, have been reported to display higher affinity towards ERβ than to ERα [90,91,92]. 8-PN is one of the very first known ERα-preferring phytoestrogens. Moreover, the ratio of ERα/ERβ abundances in target organs influences the overall action of SERMs in that tissue. In many tissues, ERβ receptors demonstrate antiproliferative actions, while ERα receptors mediate the opposite effect [93]. Therefore, an increased ratio of ERα/ERβ correlates well with high levels of cellular proliferation [94]. ERs and GPCR1 are abundantly expressed in central and peripheral tissues, reflecting the multifunctional nature of endo- and exogenous estrogens [58]. Since 8-PN has preference for ERα, we will here focus on the tissue expression of ERα as well as the potential endocrine role of ERα–8-PN complex in the contexts of energy metabolism, pituitary function and bone physiology. 

### 6.1. ERα Tissue Expression

ERα is mainly expressed in endometrium, ovarian stroma, bone, mammary gland, placenta, pancreas, skeletal muscle, white and brown adipose tissues, and various neuroendocrine areas of the brain [95,96,97]. Strikingly, ERα is abundantly present in the ventrolateral portion of the ventromedial nucleus, the arcuate nucleus and the paraventricular nuclei of the hypothalamus as well as the medial preoptic area, whereas ERβ is significantly less expressed in these locations [98,99,100,101,102,103]. ERα occurs to a variable degree in all the secretory populations of the anterior pituitary neuroendocrine cells with a higher density in lactotrophs, gonadotrophs and somatotrophs and a lower density in thyrotrophs [104]. Therefore, the activation of ERα by potent exogenous ERα agonists, such as 8-PN, may interfere with the regulation of energy metabolism, glucose homeostasis, and other endocrine milieus, as will be discussed next.

### 6.2. Energy Metabolism

#### 6.2.1. Effects of 8-PN and Related Compounds on Energy Balance

8-PN treatment in drinking water at a dose of 10 mg/L/day for 20 weeks was reported to lead to inhibited body weight gain in OVX mice when compared with the control OVX group, an effect similar to that observed in estrogen-treated OVX mice. The lower body weight gain was accompanied by food intake reduction in 8-PN-treated OXV mice [105,106]. The majority of studies of naringenin (8-PN precursor) supplementation for at least 12 weeks to obese rodent models demonstrated a reduction in body weight gain without affecting food intake [107,108,109,110]. On the other hand, naringenin stimulated the release of the anorexigenic cholecystokinin hormone from intestinal secretin tumor cell line (STC-1) in vitro (secretion of CCK 12.5-fold at 0.1 mM, 23.0-fold at 0.5 mM, and 45.9-fold at 1.0 mM compared with control) [111]. Hence, these studies show that 8-PN and its precursor influence body weight gain, but the mechanism warrants further research.

#### 6.2.2. Effects of 8-PN and Related Compounds on Lipid Metabolism

Provided in drinking water, 8-PN at the dose of 10 mg/L/day for 20 weeks ameliorated plasma lipid profile in a high-fat diet-induced type 2 diabetes mouse model. The changes triggered included reduced total cholesterol and triglyceride levels and enhanced HDL concentration and HDL/LDL ratio in plasma. In the same study, 8-PN activated AMPK signaling, thereby inhibiting SREBP-1c protein expression and its downstream lipogenic enzyme targets FAS and ACC [105]. Additionally, liver and muscle take-up fatty acids mainly through the translocase CD36 [112]. Vascular endothelial growth factor receptor-1 (VEGFR-1) and its ligand VEGF-B induce the expression of fatty acid transport proteins involved in lipid transport and uptake from blood to tissue [113]. In the afore-mentioned mouse study [105], the authors also recorded downregulation of hepatic and muscle CD36, VEGFR-1, and VEGF-B by 8-PN, which may have resulted in prevention of ectopic lipid accumulation [106].

In another study, 8-PN along with XN and IX were shown to be potential ligands for human farnesoid X receptor (FXR), based on collective findings from fluorescence titration, molecular docking studies and hydrogen deuterium exchange mass spectrometry. Since FXR activates the repressive orphan nuclear receptor and small heterodimer partner, thereby inhibiting gluconeogenesis and de novo lipogenesis, the observed anti-lipogenic and glucose-lowering (see below) effects of 8-PN might occur by a mechanism involving FXR [114].

#### 6.2.3. Effects of 8-PN and Related Compounds on Glucose Homeostasis and Insulin Sensitivity

Similar to E2, 8-PN has also been reported to exert an antidiabetic effect. It lowered plasma glucose levels and improved glucose handling in oral glucose and insulin tolerance tests in a type 2 diabetes mouse model. It was further shown to augment the protein abundance of the insulin-regulated signaling molecule AS160 in skeletal muscle, suggesting improvement of glucose uptake by this tissue [106]. Insulin stimulates the PI3K/AKT signaling cascade which activates AS160 downstream, triggering GLUT4 translocation to cell membrane [115].

Regarding humans, naringenin supplementation (150 mg t.i.d for 8 weeks) proved to reduce body weight and insulin resistance as well as upregulate PPARα and PPARγ [110].

Taking together, although there is a paucity of data available regarding the metabolic and endocrine impacts of 8-PN, this phytoestrogen appears a promising therapeutic candidate in protection against diet-induced obesity and metabolic dysfunctions. 

### 6.3. Pituitary Function 

#### 6.3.1. Effects on LH and FSH Secretion

Similar to E2 at a concentration of 10^−9^ M, 8-PN at a 1000-fold higher concentration directly suppressed LH release in a primary culture of rat pituitary cells. This response was significantly antagonized by the pure ER antagonist ICI 182,780 (Fulvestrant, Faslodex) [105]. Interestingly, high repeated oral dosing of 8-PN in diet (68.4 mg/kg/day for 12 weeks) to OVX rats also suppressed serum LH. In magnitude, serum LH was damped by 8-PN as effectively as by a treatment with 0.17 mg/kg BW/day E2, suggesting that a long-term intake of 8-PN could counteract the hot flashes [116]. Noteworthily in the same study, serum FSH concentration was lowered by a high E2 dose (0.7 mg/kg/day) alone. At the pituitary level, 8-PN and the lower E2 dose reduced the mRNA abundances of α- and β-subunits of LH and gonadotropin-releasing hormone (GnRH) receptor. ERβ mRNA was increased by 8-PN and the higher dose of E2 [116].

The lack of E2 in postmenopausal women causes changes in the release of neurotransmitters involved in the regulation of the hypothalamic GnRH pulse generator, and thereby results in overactivity of the pulse generator [117]. These affected neurotransmitters spill over into adjacent neurons involved in the regulation of temperature and heart beats, inducing hot flashes [118,119]. Therefore, LH suppression by estrogenic compounds alleviates this discomfort. Similar to what was reported in rats [116], 8-PN did not affect circulating FSH concentrations. The single doses tested (50–750 mg) were also well tolerated in postmenopausal women [37]. In the same study, the highest of these doses was shown to significantly decrease serum LH concentration in these women. This finding supports the view that the in vitro and in vivo findings recorded in rats and rat cells on 8-PN-induced inhibition of LH secretion (see above) are also relevant to humans, which forms a promising basis for the treatment of menopausal symptoms. However, this seems to require a high dose of 8-PN. Further support to the potential of 8-PN as a future remedy to hot flashes in women provides the finding that daily intake of a hop extract standardized on 8-PN (100 or 250 μg) exerted an alleviating therapeutic effect on vasomotor and other menopausal discomforts in women [120].

#### 6.3.2. Effects on Other Pituitary Hormones

E2 or 8-PN at concentrations of 10^−9^ and 10^−6^ M, respectively, directly induced TSH release by a primary culture of rat pituitary cells [105]. This response was significantly antagonized by the ER antagonist ICI 182,780. In the in vivo part of this study, a single oral dose of E2 (15.5 mg/kg BW) or 8-PN (161.4 mg/kg BW) elevated the circulating TSH level in OVX rats. Concurrently, E2 showed lowering effects on free and bound T3 while 8-PN demonstrated a declining tendency for total T3 alone. In stark contrast to the response to a single dose, a high daily dose of 8-PN (68.4 mg/kg BW/day for 12 weeks) increased circulating total T3 in OVX rats. However, neither E2 nor 8-PN at any oral doses tested affected total T4 or free T4 in OVX rats [105]. 

ERα is also expressed in growth hormone (GH)-immunoreactive cells in the pituitary, and radiolabeled estrogen can be found in these cells [121]. Circulating GH stimulates IGF-1 production in multiple tissues including the liver and bone [122], and IGF-1 acts as a downstream mediator of GH effects, exerting feedback inhibition on GH secretion [123]. 8-PN at a concentration of 0.5 μM directly suppressed GH synthesis in GH3 pituitary adenoma cells of the rat [124]. In light of this finding, it was highly surprising that a 3-month treatment with either E2 (0.7 mg/kg/day) or 8-PN (68.4 mg/kg/day) increased serum GH in OVX rats. However, this probably resulted from weakened feedback inhibition by IGF-1, whose serum concentration was decreased by both E2 and 8-PN [125]. An identical outcome was previously reported in estrogen-treated postmenopausal women [123]. 

## 7. Regulation of Bone Homeostasis

One of the common consequences of estrogen decline in menopause is loss of bone mass that may lead to osteoporosis. Similar to E2, 8-PN and naringenin were reported to promote osteogenic differentiation in vitro [126]. In order to gain further insight into the importance of the prenyl group in 8-PN for its antiosteoporotic effects, 8-PN and naringenin were compared in in vitro conditions [127]. 8-PN was found to have a stronger ability than naringenin to improve osteoblast differentiation and osteogenic function in cultured rat calvarial osteoblasts. Concomitantly, 8-PN was more effective in inhibiting osteoclastogenesis, inducing osteoclast apoptosis and reducing the resorptive activity of osteoclasts in rabbit bone marrow cells, thus confirming the importance of the prenyl group in the naringenin structure for the bone protective mechanism. In another in vitro study, 8-PN closely mimicked E2 by enhancing osteoblast activity in the MC3T3-E1 osteoblast cell line and inhibiting osteoclastic differentiation from the multinuclear macrophage RAW264.7 [128]. In this study, the effects of 8-PN could be largely abolished by the selective ERα antagonist MPP but not with the selective ERβ antagonist PHTPP. The magnitude of 8-PN effects was smaller than that of a 1000-fold lower concentration of E2, but larger compared with equimolar concentrations of genistein and daidzein.

One of the first in vivo studies on the effects of 8-PN on bone metabolism showed that in OVX rats, a 28-day sc. treatment with 8-PN at the highest dose tested (18 mg/kg/day) prevented OVX-induced trabecular bone loss. Furthermore, there was no difference between the two 8-PN enantiomers in their capabilities of inhibiting the OVX-simulated bone loss [129]. Excellent results were also reported in a subsequent study in OVX rats [130]. Here, the duration of treatment (via feed) was 12 weeks. At 68.4 mg/kg/day, 8-PN improved the biomechanical properties of bone much to the same degree as 0.7 mg/kg/day E2. 8-PN also increased cancellous bone mineral density, although not as effectively as E2. Overall, the two other phytoestrogens tested, genistein and resveratrol, proved less promising treatment alternatives for osteoporosis in this animal model.

In contrast to these positive outcomes, in another study conducted in OVX rats, daily sc. treatment with 1.77 mg/kg 8-PN for 10 weeks failed to bring about any noticeable improvement in the biomechanical properties or structure of osteoporotic lumbar vertebrae and femora, either alone or in combination with whole-body vertical vibration [131]. 

As to humans, a cross-sectional epidemiological study of some 1700 healthy women revealed that phalangeal bone ultrasound values were greater in beer drinkers compared with subjects avoiding beer drinking and/or drinking wine [132]. This association suggests that a modifying role for 8-PN in bone structure is also possible in humans, but due to the discrepant findings in rats, more studies are needed.

## 8. Effects on Tumor Cells In Vitro

A variety of in vitro studies have probed the impact of 8-PN and related hop flavonoids on cancer cell invasion, proliferation and apoptosis. In the majority of these studies, 8-PN along with its precursors and other naringenin derivates displayed growth inhibitory and apoptotic effects in various cancer cell lines. For example, Delmulle et al. (2006) exposed the human prostate cancer cell lines PC-3 and DU145 to 8-PN, 6-PN, XN, IX and desmethyl-XN for 2 days. They found inhibited growth of the cells, with the following order of compound potency (IC_50_ [µM] for DU145 and PC-3, respectively): XN (12.3, 13.2) > 6-PN (29.1, 18.4) > 8-PN (43.1, 33.5) > IX (47.4, 45.2) > desmethyl-XN (53.8, 49.9) [133]. In these cell lines, 8-PN, IX and 6-PN were shown to induce a caspase-independent form of cell death, suggested to be autophagy [134]. 8-PN and 6-PN also dose-dependently (6.25–100 µM) reduced proliferation of PC-3 and UO.31 renal carcinoma cells [135]. In the human leukemic T lymphocyte cell line Jurkat, the antiproliferative and proapoptotic effects of 8-PN were associated with inhibition of voltage-gated Kv1.3 potassium channels [136].

At concentrations of 40 and 50 μM, respectively, 8-PN and IX decreased the viability of Caco-2 cells. While the toxicity of IX was associated with a concentration-dependent increase in G2/M ratio and an increased sub-G1 cell-cycle fraction, the treatment with 8-PN was associated with an elevated G0/G1 ratio and an increased sub-G1 cell-cycle fraction [137]. In another study, female hop flavonoids suppressed cell growth and induced apoptosis in the breast cancer Sk-Br-3 cell line 24 h post-incubation with IC_50_ values of 7.1, 22.6, and 41 µM for XN, 8-PN and IX, respectively [138]. Likewise, 8-PN suppressed the proliferation of human Burkitt lymphoma cells at concentrations > 50 μM. The degree of growth inhibition was 17% at 50 μM and 41% at 100 μM [139]. In clinical oncology, histone deacetylase (HDAC) inhibitors are currently investigated as new anticancer compounds. 100 µM 8-PN and 6-PN inhibited all 11 conserved human HDACs of classes I, II, and IV. Treatment of human melanoma SK-MEL-28 cells with 8-PN or 6-PN induced hyperacetylation of the histone complex H3 within 2 h. Furthermore, these compounds imparted a prominent, dose-dependent reduction of cellular proliferation and viability of SK-MEL-28 as well as BLM melanoma cells [140]. 

Some studies have reported differential impacts of 8-PN on malign and normal cells. For example, 8-PN and 6-PN displayed cytotoxic activity against the HeLa cervical cancer cells (with IC_50_ values ranging from 10 to 60 μM), but were less toxic to normal HL-7702 cells [141]. Similarly, 8-PN exhibited a strong growth inhibitory effect on human colorectal carcinoma HCT-116 cells with an IC_50_ value of 24 μg/mL after 48 h. However, at similar concentrations and experimental timepoints, the compound did not show any cytotoxic effect to non-cancerous colon cells (CCD-41) [142]. Based on these studies, the prenyl group seems to be crucial to the anticancer activity of flavones, since it may lead to enhanced cell membrane targeting and thus increased intracellular activity [143].

In contrast to the antiproliferative and anticancer effects of 8-PN described above, some other studies have reported potentially untoward effects in cancer cell lines. For example, 8-PN at concentrations of 10^−5^ and 10^−6^ M was found to stimulate E-cadherin-dependent aggregation and growth of MCF-7/6 cells in suspension [144]. E-cadherin/β-catenin complexes are regulated by E2 at transcriptional and/or post-translational level in mammary cancer cell lines [144]. These complexes play an important role in maintaining epithelial integrity, and their aberrant expression is associated with a wide variety of human malignancies [145]. Similarly, in a 7-day colony-forming assay in the V79 fibroblast cell line, concentrations of 8-PN up to 10 µg/mL (=~30 µM) promoted cell proliferation by approximately 2-fold. At concentrations of ≥30 µg/mL all cells died [146]. Likewise, 8-PN induced cell growth in H295R/MCF-7 co-culture with an EC_50_ of 9.9E-6 mg/mL even in the presence of the aromatase inhibitors letrozole or tamoxifen. This finding suggests that the use of phytoestrogen supplements should be avoided during breast cancer treatment [147]. 

Low toxicity of 8-PN was also reported in the RAW 264.7 murine macrophage cell line, where the LC_50_ for 8-PN was 72 µM [148]. Furthermore, 8-PN at concentrations of 1–50 µM failed to cause cytotoxicity in human acute myeloid leukemia (HL-60) cells and tended to increase metabolically active cells when tested in MCF-7 breast cancer cells [149].

In estrogen-responsive cancer cell lines, the concentration of 8-PN appears to be decisive to the outcome as shown by Brunelli et al. (2007). At concentrations below 10 μM, 8-PN showed estrogenic properties, increasing the growth of ER positive MCF-7 human breast cancer cells, while higher concentrations of 8-PN inhibited proliferation and induced apoptosis [75].

## 9. Other Beneficial and Adverse Effects of 8-PN

### 9.1. Cytotoxicity to Somatic Primary Cells

Regarding human somatic primary cells, no decrease in cell viability was detected in human umbilical vein endothelial cells (HUVEC) or human aortic smooth muscle cells (HASMC) at any 8-PN concentration tested (highest 20 µM). In fact, the converse was true of HUVEC, in which 20 µM 8-PN caused a significant increase in viable cell number. Meanwhile, 10 and 20 µM concentrations of XN and IX led to reduced viability of HUVEC and/or HASMC [150]. In keeping with these findings, apoptosis was dose-dependently inhibited by 8-PN but promoted by XN and IX in both cell lines. On the other hand, cell proliferation displayed opposite effects only in HUVEC (increased by 8-PN, decreased by XN and IX). In HASMC, XN, and IX inhibited proliferation and 8-PN exhibited the same tendency, albeit weaker. Overall, the in vitro toxicity of 8-PN to somatic cells seems to be low and it exhibits a proliferative tendency.

### 9.2. Effects on Gonadal Cells

Similarly to E2, 8-PN significantly stimulated capacitation and the acrosome reaction in incapacitated epididymal mouse spermatozoa, compared with untreated controls. Unexpectedly, however, 8-PN proved to be some 1000 times more potent than E2, being active already at low nM concentrations. The SERM compound hydroxytamoxifen failed to interfere with these actions. In capacitated sperm, E2 had no discernible effect whereas 8-PN was again able to induce the acrosome reaction. Both E2 and 8-PN enhanced sperm fertilizing ability in vitro [151]. Another phytoestrogen, genistein, exhibited the same effect pattern and potency, and low concentrations of genistein and 8-PN were more effective in combination than individually to accelerate capacitation [152]. Moreover, sensitivity to genistein was even higher for human than mouse sperm (sensitivities to 8-PN were not tested). As precocious acrosome reaction may impede fertility in vivo, these findings are alarming and call for further studies, especially in vivo.

8-PN and its precursor, IX, were shown to disrupt androgen production in rat Leydig cells. These compounds inhibited hCG-stimulated androgen synthesis at all stages of Leydig cell development and reduced cAMP production. However, they failed to suppress androgen production activated by an exogenous cAMP analog in adult and immature stages of Leydig cells. These results thus suggest a cAMP-dependent cellular steroidogenesis effect of 8-PN and IX [153]. 

8-PN at concentrations of 1 and 3 µM inhibited expansion of porcine cumulus oocytes in vitro while 3.12 µM E2 did not. On the other hand, whereas that concentration of E2 diminished *Cyp19* (aromatase) gene expression by ~40% in these cells, 8-PN tended to increase it ~2-fold at all concentrations tested (0.1–3 µM; significantly at 0.3 µM). Both compounds impaired spindle formation, but the morphology of the meiotic spindles after these treatments was different [154]. Some of the recorded effects on oocytes may thus be non-ER-mediated, and 8-PN exhibits high in vitro potency in exerting them.

The developing oocytes are surrounded by granulosa cells in the ovary, and these also express aromatase. An extremely potent inhibitory impact on aromatase activity by 8-PN was reported in KGN human granulosa-like tumor cells, with an IC_50_ level of only 8 nM. Concomitantly, 8-PN (3 µM) induced *CYP19A1* gene expression about 3-fold but did not affect CYP19A1 protein abundance at 24 h, implying that 8-PN is a catalytic inhibitor of this enzyme [155]. Of note, none of the other phytoestrogens tested exhibited aromatase inhibition.

### 9.3. Effects on Aromatase in Other Cell Types

Aromatase is a major enzyme in the biosynthesis of steroids, catalyzing a critical step for estrogen production from circulating androgens. It is expressed not only in the gonads but also in adipose tissue, vasculature, bone, brain, placenta, fetal liver, and estrogen-dependent cancers [156]. In choriocarcinoma-derived JAR cells, which express high levels of aromatase, 8-PN proved to be a highly potent inhibitor of its activity. The IC_50_ was as low as 65 nM, while for XN and IX it was 20 and 140 µM, respectively. However, no effect was recorded on aromatase gene expression (*CYP19*) [157]. A similarly potent inhibitory impact on aromatase activity was reported in the aromatase-expressing breast cancer cell line Sk-Br-3, in which the IC_50_ for 8-PN was 80 nM [138]. Slightly higher IC_50_ values were obtained in H295R human adrenocortical carcinoma cells, human placental microsomes and human breast fibroblasts, 100, 200, and 300 nM, respectively [147].

### 9.4. Effects on Other Enzymes of Clinical Relevance

AKR1B1 or human aldose reductase mediates the first step in the reduction of glucose to sorbitol in the polyol pathway, which under hyperglycemic conditions is co-responsible for the diabetic complications (i.e., retinopathy, neuropathy, nephropathy, cataract) [158]. Consequently, aldose reductase inhibitors are at the focus of research aiming at the prevention of these complications. The AKR1B1 homologue AKR1B10, in turn, is a NADPH-dependent oxidoreductase that converts carbonyl group containing compounds to their corresponding alcohols. It is overexpressed in several cancers and precancerous lesions, thus possibly playing a crucial role in the development of cancer [159]. 8-PN proved to inhibit both enzymes with IC_50_ values of 0.81–1.87 and 0.99–3.96 µM for AKR1B1 and AKR1B10, respectively. IX was a roughly equipotent inhibitor with 8-PN while XN was about 10-fold less potent. None of them influenced the activity of the closely related AKR member AKR1A1 [160].

In addition to AKR1B1 and AKR1B10, in human chemoresistant cancer cell lines also carbonyl reductase 1 (CBR1) is upregulated and partially responsible for the resistance. Therefore, CBR1 inhibitors to be given adjuvantly with cytostatic therapy could be therapeutically advantageous. 8-PN consistently inhibited CBR1 activity with IC_50_ values ranging from 0.4 to 11.7 µM depending on test system and substrate. XN and IX were up to 6-fold less potent in this regard [161].

8-PN did not exhibit appreciable acetylcholinesterase inhibition, but it did inhibit plasma cholinesterase, albeit modestly (IC_50_ = 86.6 µM) [162].

### 9.5. Effects on Blood Vessels

8-PN inhibited growth factor-induced angiogenesis of bovine endothelial cells in vitro, with an IC_50_ between 3 and 10 µM. In the chicken early chorioallantois membrane, it reduced vessel diameter and tended to shorten vessel length. In both assay types, 8-PN was roughly equipotent to genistein [163].

In stark contrast to these findings, 8-PN was found to stimulate formation of capillary-like structures by primary HUVEC on GFR-Matrigel^®^, whereas XN and IX tended to inhibit it. Supporting evidence was obtained from in vivo experiments: in a mouse Matrigel^®^ plug assay, IX and (partly) XN blocked the vascularizing effect of VEGF but 8-PN did not. In rat wound-healing assay, 8-PN enhanced angiogenesis in the wounded skin tissue whilst XN and IX impaired it. However, further blurring the view were findings from extracellular matrix invasion, an essential step in the angiogenic process. 8-PN, XN, and IX inhibited the invasive capacity of HASMC in a fairly similar manner at all three concentrations tested (0.1, 1, and 10 µM). For HUVEC, IX led to the same outcome while XN did not impair invasiveness, and 8-PN did it only at 0.1 µM concentration [150].

More recently, impacts of 8-PN and XN on angiogenesis were also assessed in a mouse model of type 2 diabetes brought about by feeding on a high-fat diet. In the kidney, diabetes induced neovascularization, whereas in the cardiac left ventricle it impaired the formation of new blood vessels. 8-PN and XN counteracted these phenomena in both tissues, with XN tending to be slightly more powerful in the kidney and 8-PN in the left ventricle. Concomitantly, these compounds acted to normalize diabetes-associated changes in VEGF-A and VEGF-B levels in these tissues, with the afore-mentioned subtle potency difference discernible in VEGF-A concentrations [164]. Altogether, the data available on 8-PN’s impacts on angiogenesis are discrepant, suggesting tissue-specificity and non-monotonic dose-responses. Thus, additional studies are called for.

Prostacyclin (PGI_2_) is a critical vasculo- and cardioprotective agent [165]. A 10-nM concentration of 8-PN proved to double PGI_2_ production by HUVEC [148]. However, at higher concentrations the effect diminished, levelling off at 10 µM 8-PN. This increase could be abolished by the selective COX-2 inhibitor NS-398. The authors therefore analyzed the effect of 8-PN on COX-2 protein expression in HUVEC. By 48 h, 10 nM 8-PN increased it by 55%. This change, in turn, could be blocked by the non-selective ER antagonist ICI 182780. Hence, the induction of PGI_2_ production by 8-PN appears to be attributable to its estrogenicity.

### 9.6. Effects on Inflammation and Immune Reactions

A high, pharmacological concentration of 8-PN (30 µM) was shown to inhibit lipopolysaccharide (LPS)-induced gene expression in RAW 264.7 murine macrophage cells. A dose-response experiment revealed that already a 10-µM concentration was almost as effective as 30 µM in inhibiting the release of pro-inflammatory mediators (TNFα, NO, PGE_2_). The mechanism appeared to involve NF-κB, whose LPS-induced DNA binding was virtually abolished by 10 and 30 µM 8-PN [148].

In support of that report, 8-PN at ~7.5 or 15 µM concentrations inhibited IL-12 secretion by murine splenic macrophages stimulated by LPS or LPS + IFNγ. 8-PN was equally effective to IX but less powerful than XN [166].

Overproduction of hyaluronan may occur and be harmful in, e.g., osteoarthritis. Hyaluronan is exported from fibroblasts and chondrocytes by the ATP-binding cassette transporter multidrug resistance associated protein 5 (MRP5) [167]. Therefore, inhibitors of MRP5 might have therapeutic value. 8-PN was found to inhibit hyaluronan export from bovine chondrocytes by MRP5 with an IC_50_ of 15 µM. IX was again equipotent to it and XN twice as effective. A high concentration of 8-PN (50 µM) also inhibited proteoglycan loss and collagen degradation induced by interleukins [168].

When applied topically daily on rat skin wounds for 7 days, 50 µM 8-PN increased whereas XN and IX reduced the inflammation markers *N*-acetylglucosaminidase activity and IL1β levels in serum. This was associated with augmented formation of granulation tissue by 8-PN while XN and IX alleviated its formation [150]. The opposite effects of these compounds in this case are noteworthy even though they occurred at pharmacological doses.

### 9.7. Effects on Platelets and Blood Coagulation

Phytoestrogens appear to impart beneficial effects against cardiovascular disease, mainly by inhibiting platelet function [169]. Moreover, 8-PN was demonstrated to inhibit platelet aggregation induced by different agonists and platelet adhesion to collagen matrix. Although 8-PN activated the inhibitory NO/cGMP/PKG/VASP pathway in human platelets, this pathway was not involved in 8-PN-mediated inhibition of platelet function. Instead, the authors concluded the inhibitory mechanism to rely on different downstream molecules of collagen and thrombin receptor engagements, including Pyk2, Akt, and ERK1/2. Of note, this effect was not mediated by ERs [170]. However, in postmenopausal women, daily treatment for 5 days with a standardized extract from spent hops did not interfere with blood clotting [171]. This may be ascribable to dosing or counteracting mixture effects, and additional research on this topic is needed.

### 9.8. Effects on Muscle

In primary mouse myotubes in vitro, 8-PN at physiological concentrations activated the PI3K/Akt/P70S6K1 pathway. In vivo, dietary 8-PN accelerated muscle recovery from disuse atrophy and prevented the reduction of Akt phosphorylation in male C57BL/6 mice with immobilized right limb. Since E2 also enhanced the recovery of muscle mass, it is likely that the estrogenic activity of 8-PN was involved in its positive impact [172].

In another muscle disuse model in mice, 18-day preoperative 8-PN administration in feed reduced sciatic denervation-induced muscle atrophy. Akt phosphorylation was lowered by denervation in control mice, but 8-PN intake retained the phosphorylation status on Akt in denervated muscle comparable to that seen in sham-operated muscle. Denervation also induced the expression of atrogin-1 (a skeletal muscle-specific ubiquitin ligase), but 8-PN curtailed this increase by about 50% [173].

### 9.9. Effects on Neurons and Behavior

The generation of new neurons in the adult brain from neural stem cells (neurogenesis) offers the potential for endogenous brain repair and functional regeneration. Adult neurogenesis is partially under the control of E2, and therefore 8-PN as a potent phytoestrogen has attracted research interest in this field.

In mouse embryonic forebrain cells, 10 µM 8-PN exhibited no activity in promoting neuronal differentiation when assessed by a promoter assay after a three-day incubation period. However, after seven-day incubation, evidence of significant promotion was obtained by immunocytochemistry [174,175]. Additional studies with related compounds implied that estrogen-like activity of prenylflavanones can be dissociated from their activity of differentiation induction in neural precursor cells [175].

Using in silico prediction of molecular properties and molecular docking, Monteiro et al. (2018) [176] assessed 46 phytochemicals for their neuroprotective activity potential by virtue of binding to certain key targets for Parkinson’s and Alzheimer’s diseases, concluding that 8-PN is among the best drug candidates for both. Related to this, in contrast to four other compounds extracted from *Sophora flavescens*, 8-PN did not block the conversion of L-tyrosine to L-DOPA by tyrosinase in vitro [177].

To evaluate their potential as aids in insomnia and anxiety management, 8-PN, IX, and XN were subjected to a study on their effects on GABA_A_ receptors. Of the three compounds, 8-PN proved the most potent at displacing the non-competitive GABA_A_ receptor blocker ethynylbicycloorthobenzoate (EBOB) from native and recombinant GABA_A_ receptors, both in the presence and absence of GABA. For example, the IC_50_ values for the potentiation of GABA-induced displacement of [^3^H]EBOB in native GABA_A_ receptors were 7.3, 11.6, and 29.7 µM for 8-PN, IX, and XN, respectively. There was moderate GABA_A_ receptor subtype selectivity discernible. The GABAergic modulatory effects of these hop prenylflavonoids appeared not to be mediated via the high-affinity benzodiazepine binding site [178].

An in vivo study in rats yielded supporting evidence for behavioral modulation by 8-PN. When rats were treated with 10 mg/kg/day racemic 8-PN for 21 days, they exhibited prolonged escape latencies from the elevated T-maze open arm, implying a panicolytic effect. The positive control substance fluoxetine acted similarly. A subsequent docking study revealed that the *R* configuration of 8-PN had greater affinity to the transporters for serotonin, noradrenaline, and dopamine than does the *S* enantiomer, suggesting that the (*R*)-8-PN is the active form [179].

### 9.10. Effects on Barrier Tissue Integrity

8-PN was capable of both preventing TNFα-induced epithelial disruption in the human intestinal epithelial cell line Caco-2 and restoring barrier integrity after TNFα-induced dysfunction, whereas XN and IX were ineffective and 6-PN only effective in the prevention phase. Based on these findings, 8-PN has potential as a remedy for a variety of diseases in which defective intestinal epithelial barrier function plays a major role, such as celiac disease, Crohn’s disease, and inflammatory bowel disease [180].

TR146 cells are epithelial cells derived from human buccal carcinoma. Exposure of these cells to 5 µM 8-PN or tamoxifen increased E-cadherin protein abundance in them, and irradiation of the cells did not interfere with this change. The E-cadherin/catenin complex at the membrane of normal epithelial cells is instrumental in maintaining cell–cell adhesion [181]. When TR146 cell aggregates were irradiated in vitro, the aggregate volume declined due to cell shedding. However, 5 µM 8-PN or tamoxifen totally prevented the volume reduction. In vivo, mice treated topically on tongue mucosa with 8-PN or TAM displayed a significantly delayed onset of irradiation-induced oral mucositis and ulcers in the tongue [182]. Thus, 8-PN appears to be effective in maintaining epithelial barrier integrity also in the mouth.

### 9.11. Antimicrobial Effects

8-PN also has antimicrobial properties. It showed significant activity against methicillin-sensitive and -resistant *Staphylococcus aureus* and *Staphylococcus epidermidis* strains with MIC80 (the concentration that inhibits the growth of 80% of organisms) values of 25 or 50 μg/mL. However, it proved to be inactive against *Listeria monocytogenes* and *Salmonella typhimurium* [183]. Likewise, it failed to inhibit the growth of *Escherichia coli* [184]. On the other hand, 8-PN was equally effective against both the drug-sensitive wildtype strain and two drug-resistant strains (resistant to pentamidine or to pentamidine and melarsoprol) of the single-celled parasite *Trypanosoma brucei* with an EC_50_ of 6–7 µg/mL [185].

8-PN proved to possess notable antifungal activity as well. For *Trichophyton mentagrophytes*, it had the same minimum inhibitory concentration (MIC) as the well-established dermatophytosis drug griseofulvin (6.25 µg/mL). For *Trichophyton rubrum* and *Mucor rouxianus* the MIC was 12.5 µg/mL. 8-PN showed no activity against the yeast *Candida albicans* or the plant pathogen fungus *Fusarium oxysporum* [8].

### 9.12. Effects on Oxidative Stress

The data on the influence of 8-PN on oxidative stress are more discrepant than for most other phenomena. Soon after the discovery of 8-PN it was reported that at a 25 µM—but not at a 5 µM—concentration, 8-PN was able to inhibit oxidation of human low-density lipoprotein (LDL) induced by copper sulfate in vitro. IX and especially XN exhibited this ability already at the 5 µM concentration [186]. However, no antioxidant activity by 8-PN was detected in the 2,2-diphenyl-1-picrylhydrazyl test or after menadione treatment of HL-60 cell cultures (tested concentration: 10 µM) [149].

In contrast, pre-treatment with 8-PN reduced dose-dependently oxidative stress induced by LPS (0.1 µg/mL; 12 h) in RAW 264.7 murine macrophage cells, abolishing it fully at 10 µM concentration [148]. At the tested concentrations of 10 and 25 µM, 8-PN also displayed antioxidant activity in H_2_O_2_-treated H2P2G cells and in the 2,20-azino-bis(3-ethylbenzothiazoline)-6-sulfonic acid (ABTS) radical cation assay, respectively [175]. Moreover, in a N,N-dimethyl-*p*-phenylenediamine dihydrochloride (DMPD•^+^) scavenging assay 8-PN showed dose-dependent antioxidant activity, albeit not prominent, while naringenin was practically ineffective and 6-PN even demonstrated slight pro-oxidant effects at high concentrations [184].

In a mouse model of dietary type 2 diabetes, oxidative stress evolved in the liver and kidney as evidenced by 7- and 5-fold, respectively, augmented intrinsic fluorescence of advanced glycation end products (AGEs) in these tissues [187]. AGEs are modified proteins and/or lipids which are induced under oxidative conditions and which themselves enhance reactive oxygen species formation [188]. 8-PN treatment (0.1% in drinking water [~130 mg/kg/day]) was capable of fully preventing the AGE increase in the liver and of reducing it by approximately 50% in the kidney. XN also showed an inhibitory effect but less prominent than 8-PN. Another index of oxidative stress, protein nitration, was likewise elevated by diabetes in the liver and kidney (cortex). In this instance, 8-PN blocked the increase in both tissues whereas XN was only effective in the liver [187].

In contrast to the ABTS cation assay outcome described above, 8-PN (5–25 µM) was not able to reduce stable colored ABTS radicals in vitro in a more recent study. In the in vivo part of the same study, 8-PN at the tested 100 µM concentration slightly diminished intracellular accumulation of reactive oxygen species in *Caenorhabditis elegans*, but the inhibition did not reach statistical significance [189]. Overall, it thus appears that 8-PN has potential antioxidant activity, but this activity only emerges at relatively high concentrations.

### 9.13. Modulation of Other Compound Toxicities

Unexpectedly, 8-PN augmented the formation of aflatoxin B1-DNA adducts in primary human hepatocytes (statistically significant change at 10 µM). 8-PN also proved to be a potent inhibitor of the human cytochrome P450 enzymes tested, with IC_50_ values of 1.7, 4.5 and 8.4 for CYP1A1, CYP1A2 and CYP3A4, respectively. At the gene expression level, only the abundance of *CYP1A1* mRNA was altered (doubled by 10 µM 8-PN). The mechanism by which 8-PN enhances DNA adduct formation by aflatoxin B1 remained thus elusive [190].

## 10. Conclusions and Future Research Needs

8-PN is a potent phytoestrogen with a multitude of target organs and tissues. It also influences a wide variety of cellular signaling pathways, both ER-dependent and ER-independent (some examples are shown in Figure 2). While these effects have been repeatedly and convincingly established in vitro, the in vivo data are much more meagre. Furthermore, in most cases, the effects have required such high concentrations that they cannot be reached physiologically from dietary exposure to 8-PN and its precursors. The inter-individual differences in intestinal and hepatic generation of 8-PN from IX are yet noteworthy if hop-derived dietary supplements are used. 

Getting back to our original question posed in the title of this review, is 8-PN ultimately a friend or a foe? So far, researchers have focused on emphasizing the health-promoting impacts of 8-PN, which indeed constitute an impressive list. However, critical analysis of the literature also reveals several less-favorable effects (Table 3), which should be kept in mind, especially if 8-PN is intended to be applied in medical therapy in the future. The most urgent information needs currently include the health consequences of chronic exposures to high doses of 8-PN (heavy beer drinkers with high IX conversion capability) and—closely related to this—its effects in men. Since hops have mainly found therapeutic use in the treatment of menopausal health problems in women, the great preponderance of human studies on 8-PN have understandably confined to female subjects. However, given that 8-PN has endocrine disrupting potency in males, beer is more commonly consumed by men than women, and that sperm has proven an exceptionally sensitive target to 8-PN, men should no longer be neglected in 8-PN studies.

## Figures and Tables

**Figure 1 ijms-23-03168-f001:**
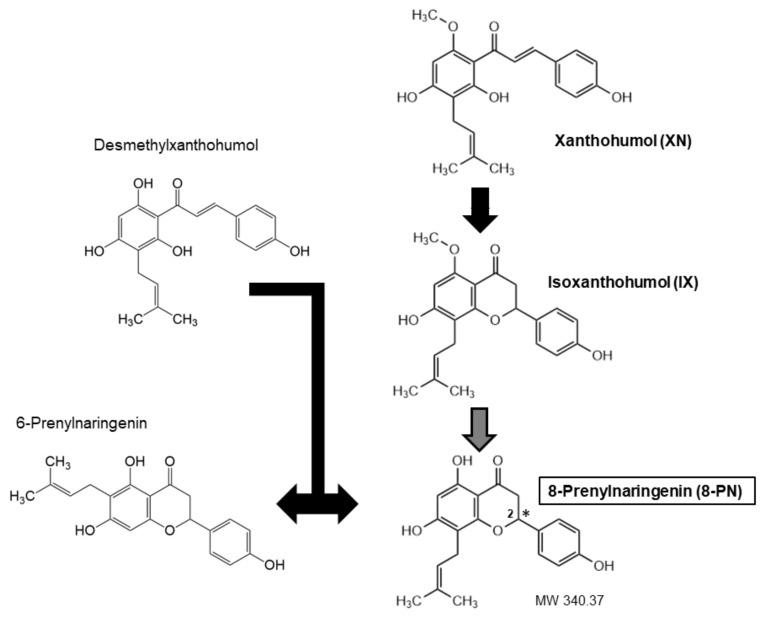
Structural formulae of prenylated flavonoids (with the most important compounds for this review in bold). XN can isomerize to IX, and desmethyl-XN to a racemic mixture of 6-PN and 8-PN. IX, in turn, may be metabolized to a variable degree to 8-PN (see text for details). The asterisk indicates the chiral center in 8-PN.

**Figure 2 ijms-23-03168-f002:**
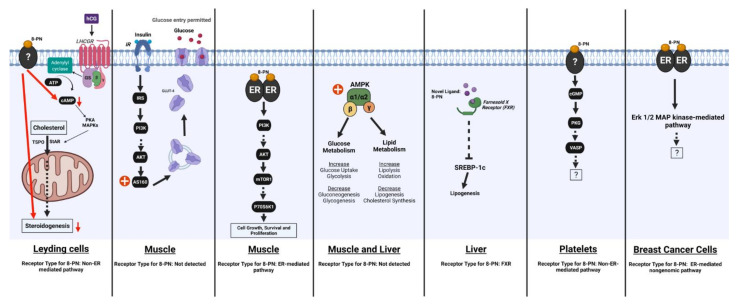
Signaling cascades mediated by 8-PN in tissues and cells (BioRender.com, license purchased). First panel: 8-PN affected steroidogenesis through cAMP-dependent pathway in primary culture of porcine Leydig cells [153]. Second panel: in mouse skeletal muscle, 8-PN stimulated the PI3K/Akt signaling pathway at AS160, which triggers GLUT4 translocation to plasma membrane [106]. Third panel: 8-PN at physiological concentrations activated the PI3K/Akt/P70S6K1 pathway in mouse myotubes and accelerated muscle recovery from disuse atrophy [172]. Fourth panel: In mice, 8-PN activated AMPK, which plays a pivotal role in lipid and glucose metabolism in muscle and liver [106]. Fifth panel: 8-PN is a potential ligand for the human farnesoid X receptor (FXR), based on collective findings from fluorescence titration, molecular docking studies and hydrogen deuterium exchange mass spectrometry [114]. Sixth panel: 8-PN directly activated the inhibitory NO/cGMP/PKG/VASP pathway in human platelets [170]. Seventh panel: 8-PN rapidly activated ERK1/2 MAP kinase in MCF-7 cells [75].

**Table 3 ijms-23-03168-t003:** Potentially beneficial and adverse effects of 8-PN.

Beneficial	Adverse
- Antidiabetic	- Affects sperm with extraordinary potency
- Counteracts the metabolic syndrome	- Impairs spindle formation in oocytes and inhibits cumulus expansion
- Relieves hot flashes	- Disrupts androgen production in Leydig cells
- Inhibits the growth of many cancer cell lines	- Promotes the growth of some cancer cell lines
- Induces osteogenesis	- Aggravates inflammatory response topically in skin
- Inhibits AKR1B1 and AKR1B10	- Inhibits transporter proteins (ABCG2, ABCB1/P-gp, ABCC1/MRP1)
- Inhibits CBR1 activity	- Alters TSH, T3, GH and IGF-1 serum levels
- Induces PGI2 production	- Inhibits aromatase in various cell types
- Mitigates LPS-induced effects	- Enhances DNA adduct formation by aflatoxin
- Inhibits MRP5	
- Inhibits platelet aggregation	
- Accelerates muscle recovery from disuse atrophy	
- Maintains barrier tissue integrity	
- Has antimicrobial properties	
- Promotes neuronal differentiation	
- Has high potential for neuroprotective activity	

## Data Availability

Only published and thus publicly available data were included.

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
