# Peer review of "The Potent Phytoestrogen 8-Prenylnaringenin: A Friend or a Foe?"

_ijms, 2022, doi:10.3390/ijms23063168_

Round 1
Reviewer 1 Report
Fine review with some formatting or orthographical mistakes, see part 6.2.2 letter size, line distance. Page 20 see 8-PM should be 8-PN (two times wrong). Page 25 section 9.12 distances wording. Page 22 is mainly empty. This are small things. Literature: several times not all authors are mentioned, instead we can find "et al.", for example No. 72, 68, 65, etc. I feel this cannot be tolerated, since all authors have contributed. The issue of optical activity: Also during cyclisation of XN it is possible that two enantiomers of IX are formed, which react in turn to two enantiomers of 8-PN. Although 6-PN is not a phytoestrogen, in Fig. 1 its formation should be shown better as a basic possibility during DXN cyclisation, there is space enough. Regarding the possibility of different activities of R and S 8-PN, I feel that in organsims there is so much biochemistry that also the repeated back reactions or ring opening/ reclosing should be discussed, so after some time there will be some racemic mixture inside the cell. Page 2, Ref. 15 citing 14,9 and 15: this here is misleading, since it is a synthesis work, better 21. Perhaps control numbers of citations.
Author Response
We want to thank the reviewers for their encouraging words and excellent suggestions to improve our review. Below, we show in detail how we have responded to each comment made by the reviewers. The reviewers’ remarks are in bold.
Reviewer 1
Fine review with some formatting or orthographical mistakes, see part 6.2.2 letter size, line distance.
These typos were rectified.
Page 20 see 8-PM should be 8-PN (two times wrong).
“8-PM” has now been replaced with “8-PN”.
Page 25 section 9.12 distances wording.
The problem with uneven spaces between words and lines in this section results from the long words and special characters in it [“2,20-azinobis-(3-ethylbenzthiazolin)-6-sulfonic acid (ABTS) radical cation assay, respectively [175]. Moreover, in a N,N-dimethyl-p-phenylenediamine dihydrochloride (DMPD•+) scavenging assay...”]. Unfortunately, there is little we can do to it because in our opinion, for clarity it is preferable not to chop the chemical names or alter the conventional symbols.
Page 22 is mainly empty.
In the revised version, we were able to correct the same problem on page 4 but unfortunately not on page 19. It appeared as if the latter page was somehow “locked” against text pasting. We therefore kindly ask the help of technical editors of IJMS in this matter.
Literature: several times not all authors are mentioned, instead we can find "et al.", for example No. 72, 68, 65, etc. I feel this cannot be tolerated, since all authors have contributed.
We fully agree with the reviewer and apologize for our negligence regarding the reference style. This has now been corrected.
The issue of optical activity: Also during cyclisation of XN it is possible that two enantiomers of IX are formed, which react in turn to two enantiomers of 8-PN.
This is true and we are grateful to the reviewer for pointing this out. We added the following sentence on page 2: “Moreover, the cyclization of XN to IX may generate two enantiomers of IX which can lead to two enantiomers of 8-PN”. We further included a citation (Kodama et al., 2007).
Although 6-PN is not a phytoestrogen, in Fig. 1 its formation should be shown better as a basic possibility during DXN cyclisation, there is space enough.
Fig. 1 was revised as suggested to show the generation of 6-PN in addition to 8-PN from DXN.
Regarding the possibility of different activities of R and S 8-PN, I feel that in organsims there is so much biochemistry that also the repeated back reactions or ring opening/ reclosing should be discussed, so after some time there will be some racemic mixture inside the cell.
We are thankful for the reviewer for this important comment. We incorporated the following sentence at the end of the chapter on 8-PN pharmacokinetics (page 7): “On the other hand, it should also be noted that back conversion to a racemic mixture might occur from both enantiomers”. This was accompanied by a citation to Caccamese et al. (2003).
Page 2, Ref. 15 citing 14,9 and 15: this here is misleading, since it is a synthesis work, better 21.
We made this change.
Perhaps control numbers of citations.
In the first section, we stated that “This review therefore seeks to present a comprehensive and up-to-date view on 8-PN”. With due respect, we do not believe that for a comprehensive review ~200 references are too much. Actually, Reviewer 2 asked us to add a couple more.
Reviewer 2 Report
This review is very comprehensive and well-organized. This referee did not find significant modifications to suggest. However, there are two papers that might be considered for citation:
https://www.hindawi.com/journals/jnme/2016/6893137/
https://www.ncbi.nlm.nih.gov/pmc/articles/PMC6017581/
The summarizing tables are highly appreciated.
Author Response
We want to thank the reviewers for their encouraging words and excellent suggestions to improve our review. Below, we show in detail how we have responded to each comment made by the reviewers. The reviewers’ remarks are in bold.
Reviewer 2
This review is very comprehensive and well-organized. This referee did not find significant modifications to suggest.
We welcome and appreciate this comment!
However, there are two papers that might be considered for citation:
https://www.hindawi.com/journals/jnme/2016/6893137/
https://www.ncbi.nlm.nih.gov/pmc/articles/PMC6017581/
We carefully considered both papers and decided to incorporate the following paragraph in the section dealing with bone: “In contrast to these promising outcomes, in another study conducted in OVX rats, daily sc. treatment with 1.77 mg/kg 8-PN for 10 weeks failed to bring about any noticeable improvement in the biomechanical properties or structure of osteoporotic lumbar vertebrae and femora, either alone or in combination with whole-body vertical vibration.” We also added the citation (Hoffmann et al., 2016).
The summarizing tables are highly appreciated.
We are happy to hear that!